# Somatization symptomology and its association with stress in patients with irritable bowel syndrome

**Saira Akhlaq**[1]*, **Nosheen Kazmi**[2], **Syed Murtaza Hassan Kazmi**[3], **Muslim Atiq**[4], **Sajawal Hussain**[5], **Ahmed Bajwa**[5], **Abdul Ahad**[5], **Muhammad Yaqoob Akhtar**[5], **Kalsoom Akhlaq**[6], **Mohammad Rizwan**[5]

**1** Shifa School of Health Professions Education, Shifa Tameer-e-Millat University, Islamabad, Pakistan, **2** Department of Psychiatry, Shifa International Hospital, Islamabad, Pakistan, **3** Department of Pulmonology, Shifa International Hospital, Islamabad, Pakistan, **4** Department of Gastroenterology, Shifa International Hospital, Islamabad, Pakistan, **5** Shifa College of Medicine, Shifa Tameer-e-Millat University, Islamabad, Pakistan, **6** Department of Internal Medicine, Shifa International Hospital, Islamabad, Pakistan

* saira.sshpe@stmu.edu.pk

## Abstract

Irritable Bowel Syndrome (IBS) patients and Somatization Symptom Disorder (SSD) patients experience somatization symptoms relative to their corresponding processes. IBS patients may also have a diagnosis of both IBS and SSD. Somatization symptoms cause significant psychological, emotional and social distress. Conversely, stress in any form is believed to contribute to IBS symptoms. Whether stress mediated somatization symptoms in patients with IBS provide a pathway for these IBS symptoms is not as well understood. This cross-sectional study was performed at Shifa International Hospital, Islamabad between March 1st, 2023, and January 14th, 2024. Purposeful sampling was done to recruit study participants from three different populations as somatization is common in all three populations. As a result, there were three different samples in the study. Participants were eligible to participate if they had a diagnosis of IBS, somatic symptom disorder (SSD), or IBS with somatization (IBS-SSD) and were currently receiving treatment at the gastroenterology outpatient clinic and/or psychiatric outpatient clinic. Patient Health Questionnaire (PHQ-15) and Somatic Stress Response Scale (SSRS) were used to assess somatic symptoms and their association of stress-related somatic symptoms. Data was entered and analyzed using descriptive and inferential statistics. Data was self-reported by the participants. The largest sample size 67(100%) was from the IBS patient population. Two other samples were small i.e., there were 21 (100%) participants in SSD sample, and a very few numbers of participants 12 (100%) in the IBS diagnosis with a comorbidity of SSD sample. Majority of the patients were young i.e., 50≤ (77.7%), (71.4%), (74.99%); and male (59.7%), (66.6%), (50.0%) from the IBS, SSD, and IBS-SSD samples. Majority of the participants in the IBS (56.7%) and SSD (61.9%) samples had a high school diploma or the equivalent. In the IBS-SSD sample, the largest percentage (41.7%) of participants had more than a bachelor's degrees. M = 85.67 (+/-23.26) for SSRS scores and M = 17.81(+/-5.28) for PHQ-15 scores in SSD patients. M = 75.21 (+/-19.59) for SSRS scores and M = 14.76 (+/-5.07) for PHQ-15 scores in IBS patients. M = 75.17 (+/-20.55) for SSRS scores and M = 14.92 (+/-6.27) for

**Data Availability Statement:** All relevant data are within the manuscript and its Supporting Information files.

**Funding:** This study was financially supported by Shifa Tameer-e-Millat University Internal Reseach Grants Committee in the form of a grant (025-2023) received by SA, NK, and MA. No additional external funding was received for this study.

**Competing interests:** The authors have declared no competing interests exist.

PHQ-15 scores in IBS-SSD patients. Many participants had somatization symptoms in the severe range ($\geq$ 15) i.e., 34(50.7%), 17(81.0%), 6(50.0%) in IBS, SSD, and IBS-SSD samples respectively. Considering the PHQ scores by age in the IBS sample, highest mean scores were observed for the highest age group (60–69 years) i.e., 16.50 (+/- 5.68) despite fewer number of participants in this age group. PHQ scores also significantly differed by education groups i.e., significant differences were observed between education group 1 and 2 as well as group 2 and 3, p<0.05. On simple linear regression, PHQ-15 scores significantly predicted variations in SSRS scores, p <0.05, $R^2$ = 69.6% for IBS sample, $R^2$ = 68.7% for the SSD sample, and $R^2$ = 66.0% for patients with IBS, SSD and IBS with somatization respectively. Stress related somatic symptoms are positively correlated with somatization complaints in IBS patients. Increased somatization scores were observed in the elderly. Targeted psycho-social interventions could help mitigate the negative effects of somatization in IBS patients.

## Introduction

Recognition of IBS patients with stress-related somatization symptoms is needed for facilitating implementation of more effective approaches for IBS. This will help to decrease the disproportionately high healthcare costs associated with IBS [1]. Stress is an important predictor variable in IBS management because the magnitude of symptom intensity in IBS patients can be predicted by measuring stress [2]. Considering stress as a psychological variable, it is unclear if the natural history of patients with IBS can be changed with psychological therapies as earlier interventions, or if a combination of psychological therapy with a central neuromodulator can have an additive effect in the treatment management of IBS [3]. In addition to stress, somatization is also a psychological variable that is directly associated with IBS severity [4]. Dysfunction in the hypothalamic-pituitary-adrenal (HPA) axis that is essential for controlling physiological stress responses may be related with somatization [5]. Considering the pathophysiology of IBS, dysregulated interaction involving gut-brain axis, leading to dysmotility visceral hypersensitivity, and altered CNS processing is an established standard [3]. Whether stress is acute or chronic, both involve mechanisms in the central nervous system (CNS) for responding to stress [6]. The role of CNS is important in considering stress-mediated responses of patients with IBS as these responses may be more prominent in IBS patients who may have greater reactivity to stress compared with healthy individuals [7]. Exaggeration of the neuroendocrine response and visceral perceptual alterations corresponding to stress may explain some of the stress related gastrointestinal symptoms in IBS [8]. Magnitude of visceral sensations, subjective emotional responses and heart rate increase in IBS patients exposed to stress as compared to healthy controls exposed to the same stressor. Thus, stress-induced modulation of visceral perception in IBS patients is altered [9]. Considering the IBS patients, more than the type of stressor, stress score itself and the response to stress holds more significance in increasing greater likelihood of negative affect in IBS patients as compared to controls. Self-reported data is more useful in differentiating IBS patients with increased likelihood of negative affect from healthy controls. In comparison, differentiation cannot be made when pattern of changes in sympathetic activation after the mental stressor are considered [10]. Thus, psychological response to stress holds more significance as compared to physiological response to stress in IBS patients. Considering the HPA responses differently to physiological

and psychological stress [6], current study is focused more on the body's response in terms of somatization symptoms in IBS patients by measuring the scores on the Patient Health Questionnaire-15 (PHQ-15) and the increase/decrease in those scores with the presence/absence psychological stress related somatization symptoms as measured through the Somatic Stress Response Scale (SSRS).

Psychosomatic disorders and functional disorders such as IBS are common presentations in general medical practice and in specialty practice. They cause clinical problems for practitioners due to their uncertain nature and lack of effective treatment [11]. Somatization is common in both populations i.e., patients with psychosomatic disorders and patients with IBS diagnosis. The trait characteristic "somatization" is defined as "a tendency to experience and communicate psychological distress in the form of somatic symptoms and to seek medical help for them" [12]. Considering symptomology in patients with a diagnosis of Somatic Stress Disorder (SSD), high somatic illness attributions despite contradicting medical information and low symptom tolerance have been observed [13]. Thus, somatization that is considered as a transient phenomenon as it is not worrisome for short period of time yet needs to be addressed if prolonged over a long period of time. Therefore, it is needed to consider the severity of somatization symptoms in any group for which somatization is being analyzed. IBS patients score higher on somatization than healthy controls, but lower than patients with somatoform disorders. Moreover, somatization is a significant psychological factor directly associated with IBS severity [4, 14]. As expected, IBS is known to have a considerable symptom overlap with other functional somatic syndromes like chronic fatigue syndrome or fibromyalgia syndrome. IBS is largely synonymous with the concepts of somatoform disorders. Roughly half of IBS patients complain of gastrointestinal symptoms only and have no psychiatric comorbidity [15]. Catastrophizing and Somatization are associated with IBS severity. Psychological distress as compared to GI symptoms has a stronger direct effect on health-related quality of life in IBS patients [16]. The negative effects of psychological factors on IBS can be decreased by reducing catastrophizing and somatization [4], especially when somatization is common in IBS patients. Patients with IBS-in comparison to those with functional constipation or functional diarrhea, have been shown to have abnormally high somatization, and clinically abnormal levels of anxiety [17]. Anxiety has an indirect effect on IBS symptoms through catastrophizing. Anxiety, in turn, was predicted by neuroticism and stressful life events [4]. Moreover, chronic life stress threat has been identified as a powerful predictor of subsequent symptom intensity in IBS patients with significantly reduced chances of clinical improvement in patients exposed to even one chronic highly threatening stressor [2]. As clinicians, however, we do realize that it's not the mere exposure to stressful life events, but a continuous "heightened stress temperature", which needs to be best measured on a continuum as a "fight and flight response". Only a structured instrument that is designed to measure stress related body somatic response can help evaluate the indirect effect of continuous stress and its related effect on somatic symptoms. This will help target strategies for patients in the somatoform spectrum, including those with IBS. There is an unmet need for targeted approaches to manage somatization symptoms, especially stress-related somatic symptoms in addition to the biomedical approaches for the treatment of somatization symptoms in patients with IBS and SSD. The presence of psychiatric and somatic co-morbidities has been reported in IBS patients [18]. Therefore, some of the IBS patients may have SSD as a comorbid condition. With this background, we conducted this study to understand the relationship between stress related somatic symptom burden and somatization symptoms in patients mainly with IBS, only a small percentage of patients with SSD diagnosis and even smaller sample size of patients with IBS-SSD diagnosis.

## Materials and methods

The cross-sectional study was performed at Shifa International Hospital, Islamabad by administering questionnaires in-person. Questionnaires were explained to remove any confusions about the information asked through the questionnaires. This also provided face-validity for the administered questionnaires. An institutional review board (IRB) approval granted through the IRBs of Shifa Tameer-e-Millat University with an IRB # 020–23. Research team members collected data by regularly following up with the patients after their clinical visits from March 1st, 2023, to January 14th, 2024, after getting verbal informed consent. Study was explained verbally and in writing on the informed consent form. Filling up the questionnaires implied informed consent to participate in the study.

### Definitions

Irritable bowel syndrome was defined by the Rome IV criteria, as recurrent abdominal pain associated with altered stool form or frequency [19]. Patients were recruited from a gastroenterology clinic. Somatic Symptoms Disorder was defined based on DSM V criteria as distressing symptoms pertaining to excessive thoughts related to health concerns over a span of at least 6 months [20]. These patients were recruited from the gastroenterology clinic for IBS or IBS-SSD diagnosis, or psychiatry clinic for SSD diagnosis.

### Inclusion and exclusion criteria

Purposeful sampling was done to recruit study participants as the purpose was to study the patterns of somatization as well as stress-related somatization scores in patients visiting these specific clinics. Participants were eligible to participate if they had a diagnosis of IBS, somatization, or IBS with somatization and were currently receiving treatment. Patients were screened for the study using two screening questions based on the eligibility criteria: "What is the type of your diagnosis?" and "How long have you been diagnosed? Patients who met the eligibility criteria i.e., a diagnosis of, IBS, IBS with somatization, or somatization were included in the study. No potential participants meeting the eligibility were excluded for any reason, unless they refused to provide voluntary consent for the study.

### Sample size calculation

G*Power (2017) [21], was used to calculate estimated sample sizes for all the samples. A sample size of 55 study participants was needed for generalizability to larger populations when conducting one-way ANOVA for estimating significance of differences in PHQ scores by education. Sample size calculation for ANOVA was done by using a moderate effect size ($f = 0.5$) with alpha set at 0.05 and power at 0.80. For linear bivariate regression, keeping the slope H1 = 0.5, slope H0 = 0, alpha = 0.05, and power = 0.80, estimated sample size needed for generalizability was 21. Thus, the required sample size was met for the IBS sample only as 67 participants completed the study- a number greater than 55 that was required for generalizability. Thus, study findings of IBS sample may be generalized to larger population. Additionally, SSD despite being small in numbers of participants may still be considered for generalizability as the number of participants were 21- a number needed for generalizability.

### Survey design

Demographic variables were selected based on the literature review. Demographic data measurement included assessing age, gender, education, marital status, and employment status. The study was meant to assess the somatic stress response in IBS patients for which 2

instruments were utilized. PHQ-15 Instrument is used to measure somatization symptom burden in patients [22]. Scoring of PHQ follows a universal standard where $\geq 5$, $\geq 10$, $\geq 15$ represent mild, moderate, and severe levels of somatization. Therefore, in the current study, a score of 1–5 was coded as participants being in category 1 which means mild symptoms. A score of 5–14 was coded as category 2 which means that participants in this category have moderate symptoms. A score of 15–30 was coded as category 3 and the participants in this category had severe symptoms. PHQ has good reliability in the IBS sample. Cronbach's alpha was 0.78.

SSRS is an instrument that measures stress-related somatic symptoms in IBS patients [23]. SSRS is a 32-item scale that is valid and reliable. The items on the SSRS are also 5-point Likert items that are added within the five subscales to produce five different variables corresponding to the types of response to stress in somatic symptom disorder (SSD). The five types of subscales include cardiorespiratory response, somatic sensitivity, gastrointestinal response, general somatic response, and genitourinary response. The instrument is reliable instrument with Cronbach's alphas ranging from .72 to .92 for each of the five subscales, and .95 for the total score. SSRS has a very high reliability. In the IBS sample, SSRS had a reliability of 0.91 as measured through the Cronbach's alpha. Convergent validity of the SSRS was calculated by correlating the SSRC scale scores with the somatization sub-scale scores and other sub-scale scores in the Korean version of the SCL-90. and Discriminant validity of the SSRS was calculated by comparing the sub-scales of the healthy study participants with the patient study participants [23]. Maximum scoring possible on the instrument was 160, minimum was 1. Participants who reported stress as "not at all" and "somewhat" was considered below average somatization symptoms due to stress, "moderately" was considered as average, "very much" and "absolutely" were considered as above average. Thus, a score from 1–64 was considered below average, a score of 64 as average, and a score of 65–160 as above average.

## Data collection

Data were collected from the eligible participants after obtaining verbal informed consent. Verbal agreement to participate in the study implied voluntary informed consent to participate in the study. Survey data was recorded on reliable and validated questionnaires for the participants who agreed to fill the questionnaires either before or after their scheduled clinical visit. Data collection took about 30–45 minutes per participant and 11 months for the overall process. Some of the patients provided incomplete data and were therefore, not included in the inferential analysis. Some of the patients refused to voluntarily participate in the study or some decided to withdraw from the study during the data collection process and were therefore, excluded from the study. Data accuracy was ensured by explaining each question to each study participant. All questionnaires had simple easy-to-read questions. The overall data collection process took about 11 months. Anonymity of the data collection process and data storage security was maintained.

## Statistical analysis

The scores for each item on all the scales, except the demographic questionnaire were individually added together for each scale to form continuous variables for calculating continuous scores. To meet research objectives, a mean score for each item was first calculated and then the average mean score for each participant was calculated.

To assess the test of normality, Kolmogorov-Smirnov test was performed. The results were significant which means that the assumption of normality was not met, yet the distribution of scores in the boxplot was around the central line. Therefore, the distribution can be considered

as a normal distribution and parametric tests can be applied. Statistical Package for the Social Sciences (SPSS) was used to secure an electronic database along with data analysis.

## Results

Data was collected for three different samples. The largest set of data was collected for the patients with IBS diagnosis. Three different samples belong to three different populations. Therefore, each group is homogenous. However, three samples have been selected because of a common issue i.e., somatization symptomology in all three samples. Despite the commonality, these diagnoses require a set of different parameters for making a diagnosis. Therefore, the three samples have been kept separate and analyzed independently for making meaning of relationships associated with the somatization symptomology in each unique sample. The sample with a diagnosis of IBS included 67 participants (100%). This was the largest sample size that also met the generalizability to larger populations. The sample of patients with a diagnosis of somatization symptom disorder (SSD) had 21 (100%) participants. Many patients belonging to this group, approximately around 30 patients who were approached for study enrollment refused to participate in the study. Therefore, no more enrollment was possible after consistent refusals. A small number of patients i.e., 12 (100%) in a sample of patients reported a diagnosis of IBS along with an SSD diagnosis. Considering the three samples from three different populations, IBS sample is the most relevant for generalizability.

Considering the three samples, majority of the participants were less than 50 years old (77.7%), (71.4%), (74.99%); male (59.7%), (66.6%), (50.0%); and married (80.6%), (76.2%), (75%) from the IBS, SSD, and IBS-SSD samples. Majority of the participants in the IBS (56.7%) and SSD (61.9%) samples had a high school diploma or the equivalent. In the IBS-SSD sample, the largest percentage (41.7%) of participants had more than a bachelor's degrees. Considering time since diagnosis most of the participants had a diagnosis for 3 years or more, in the IBS sample (37.3%), SSD sample (52.4%), and the IBS-SSD sample (33.3%) for SSD and (50.0%) for IBS. More than 50% of participants in the IBS sample (50.7%) were employed. More than 50% of the participants in the SSD (52.4%) and IBS-SSD (75.0%) samples were unemployed Table 1.

### Somatization symptom burden and stress-related somatization burden

Mild somatization symptoms i.e., 3(4.5%), 1 (4.8%), 0(0%); moderate symptoms i.e., 30 (44.8%), 3(14.3%), 6(50.0%); and severe symptoms i.e., 34(50.7%), 17(81.0%), 6(50.0%) were reported by participants from the three samples i.e., IBS, SSD, and IBS with somatization patients' samples respectively. Mean PHQ scores were recorded as 14.76 +/- 5.07 (4.0–24.0), 17.8 +/- 5.27 (5.0–26.0) and 14.92 +/- 6.27 (7.0–26.0) for patients with a diagnosis of IBS, SSD and IBS with somatization respectively.

Considering the severity spectrum on the SSRS scale, 47(70.15%) and 18(85.71%) had severe symptoms of stress-related somatization symptoms in the IBS and SSD samples. 19 (28.36%), 3 (14.29%) and had mild symptoms in the IBS and SSD samples, and 1 (1.49%), 0 (0%) had moderate symptoms in the IBS and SSD samples respectively. Mean SSRS scores were recorded as 75.20 +/-19.59 (46–136), 85.67+/-23.25 (44–127) and 75.17+/-20.55 (52–131) for patients with IBS, SSD and IBS with somatization respectively.

### Somatization symptoms in IBS patients by age and education

In the IBS patients, the highest PHQ scores and the SSRS scores (83.17 +/-23.92) were noted to be in the age group 60–69 years S1 Table. Means of SSRS scores were calculated to assess which age group in IBS patients had the highest mean score. Age 60–69 years had the highest

**Table 1. Descriptive statistics of IBS sample, SSD sample, and IBS-SSD sample.**

| Variables | Categories | Ranges/Types | Frequency | | |
|---|---|---|---|---|---|
| | | | (N) (P) (N) (P) (N) (P) | | |
| | | | IBS | SSD | IBS-SSD |
| Age | 1 | 18–29 | 19 (28.4) | 7 (33.3) | 4 (33.33) |
| | 2 | 30–39 | 15 (22.4) | 5 (23.8) | 5 (41.66) |
| | 3 | 40–49 | 18 (26.9) | 3 (14.3) | 0 (0) |
| | 4 | 50–59 | 9 (13.4) | 3 (14.3) | 1 (8.33) |
| | 5 | 60–69 | 6 (9.0) | 3 (14.3) | 2 (16.66) |
| Gender | 1 | Male | 40 (59.7) | 14 (66.6) | 6 (50.0) |
| | 2 | Female | 27 (40.3) | 7 (33.3) | 6 (50.0) |
| Education | 1 | High School Diploma or equivalent | 38 (56.7) | 13 (61.9) | 3 (25.0) |
| | 2 | Bachelor's degree | 17 (25.4) | 2 (9.5) | 4 (33.3) |
| | 3 | Higher than bachelor | 12 (17.9) | 6 (28.6) | 5 (41.7) |
| Time since diagnosis | 1 (IBS) | Less than 6 months | 15 (22.4) | | 1 (8.3) |
| | (SSD) | | | 2 (9.5) | 2 (16.7) |
| | 2 (IBS) | 6 months-Less than 12 months | 5 (7.5) | | 1 (8.3) |
| | (SSD) | | | 4 (19.0) | 1 (8.3) |
| | 3 (IBS) | 12 months-less than 2 years | 11 (16.4) | | 2 (16.7) |
| | (SSD) | | | 1 (4.8) | 3 (25) |
| | 4 (IBS) | 2 years-less than 3 years | 11 (16.4) | | 2 (16.7) |
| | (SSD) | | | 3 (14.3) | 2 (16.7) |
| | 5 (IBS) | 3 years or more | 25 (37.3) | | 6 (50.0) |
| | (SSD) | | | 11 (52.4) | 4 (33.3) |
| Marital status | 1 | Married | 54 (80.6) | 16 (76.2) | 9 (75) |
| | 2 | Un Married | 13 (19.40) | 5 (23.8) | 3 (25) |
| | Total | | 67 (100) | 21 (100) | |
| Employment | 1 | Un Employed | 33 (49.3) | 11 (52.4) | 9 (75) |
| | 2 | Employed | 34 (50.7) | 10 (47.6) | 3 (25) |
| | Total | | 67 (100) | 21 (100) | 12 (100) |

mean score. However, these differences were not significant in the ANOVA analysis. Mean scores for PHQ were significantly lower for patients with a bachelor's degree (11.71+/-5.35 vs. 15+/-4.03 and 16.7+/-6.10 for those with a high school diploma or a post graduate degree respectively). These differences were significant in the ANOVA analysis Tables 2–4.

## Relationship of PHQ scores with SSRS scores in IBS, SSD, IBS-SSD patients

On simple linear regression, variations in SSRS scores were significantly predicted by variations in PHQ scores, $p < 0.01$, $R^2 = 69.6\%$ for IBS sample, $R^2 = 68.7\%$, $R^2 = 66.0\%$ for the IBS,

**Table 2. Descriptives: PHQ scores by education in IBS patients.**

| Categories | N | Mean | Std. Dev | Std. Error | 95% Confidence Interval for Mean | |
|---|---|---|---|---|---|---|
| | | | | | Lower Bound | Upper Bound |
| High School Diploma or equivalent | 38 | 15.50 | 4.03 | .653 | 14.18 | 16.82 |
| Bachelor's Degree | 17 | 11.71 | 5.35 | 1.297 | 8.96 | 14.46 |
| Higher than bachelor's degree | 12 | 16.75 | 6.10 | 1.763 | 12.87 | 20.63 |
| Total | 67 | 14.76 | 5.07 | .619 | 13.53 | 16.00 |

**Table 3. One-way ANOVA for PHQ scores in IBS by education.**

| | Sum of Squares | df | Mean Square | F | Sig. | 95% CI Lower Upper |
|---|---|---|---|---|---|---|
| Between Groups | 226.900 | 2 | 113.450 | 4.948 | .010 | .009 .275 |
| Within Groups | 1467.279 | 64 | 22.926 | | | |
| Total | 1694.179 | 66 | | | | |

**Table 4. Multiple comparisons for PHQ scores in IBS patients by education.**

| | (I) IBS Education | (J) IBS Education | Mean Difference (I-J) | Std. Error | Sig. | 95% Confidence Interval Lower Bound | Upper Bound |
|---|---|---|---|---|---|---|---|
| Bonferroni | 1 | 2 | 3.79 | 1.397 | .025 | .36 | 7.23 |
| | | 3 | -1.25 | 1.586 | 1.000 | -5.15 | 2.65 |
| | 2 | 1 | -3.79 | 1.397 | .025 | -7.23 | -.36 |
| | | 3 | -5.04 | 1.805 | .021 | -9.48 | -.6057 |
| | 3 | 1 | 1.25 | 1.56 | 1.00 | -2.65 | 5.15 |
| | | 2 | 5.04 | 1.81 | .021 | .61 | 9.48 |

**Table 5. Linear regression for SSRS and PHQ scores in IBS patients.**

| Model | | Unstandardized B | Coefficients Std. Error | Standardized Coefficients Beta | t | Sig. | 95% CI Lower Bound | Upper Bound |
|---|---|---|---|---|---|---|---|---|
| 1 | (Constant) | 27.573 | 4.121 | | 6.691 | < .001 | 19.34 | 35.80 |
| | PHQ-15 | 3.227 | .264 | .835 | 12.213 | < .001 | 2.70 | 3.76 |

a. Predictors: (Constant), PHQ Scores in IBS patients
b. Dependent Variable: SSRS Scores in IBS patients

SSD, and IBS-SSD samples—a large effect size for the overall model in all the samples. The slope coefficient (B) of SSRS was significantly different from zero in the model indicating that there was linear relationship of SSRS with PHQ in both the samples. SSRS scores increase by 12.2 points with increasing PHQ scores, p < .001 (2.70–3.76) for IBS sample. SSRS scores increase by 6.46 points with increasing PHQ scores, p < .001 (2.47–4.84) for SSD sample. SSRS scores increase by 4.41 points with increasing PHQ scores, p < .001 (1.32–4.01) for IBS-SSD sample Tables 5–7.

**Table 6. Linear regression for SSRS and PHQ scores in SSD patients.**

| Model | | Unstandardized B | Coefficients Std. Error | Standardized Coefficients Beta | t | Sig. | 95% CI Lower Bound | Upper Bound |
|---|---|---|---|---|---|---|---|---|
| 1 | (Constant) | 20.606 | 10.478 | | 1.967 | < .064 | -1.33 | 42.54 |
| | PHQ-15 | 3.653 | .565 | .829 | 6.464 | < .001 | 2.47 | 4.84 |

a. Predictors: (Constant), PHQ Scores in SSD patients
b. Dependent Variable: SSRS Scores in SSD patients

**Table 7. Linear regression for SSRS and PHQ scores in IBS-SSD patients.**

| Model | | Unstandardized B | Coefficients Std. Error | Standardized Coefficients Beta | t | Sig. | 95% CI | |
|---|---|---|---|---|---|---|---|---|
| | | | | | | | Lower Bound | Upper Bound |
| 1 | (Constant) | 35.467 | 9.706 | | 3.654 | .004 | 13.84 | 57.09 |
| | PHQ-15 | 2.661 | .604 | .813 | 4.409 | .001 | 1.316 | 4.01 |

a. Predictors: (Constant), PHQ Scores in IBS-SSD patients

b. Dependent Variable: SSRS Scores in IBS-SSD patients

## Discussion

Severe somatization symptoms were observed in a large percentage of all three samples belonging to different populations i.e., 50.7% (IBS), 81.0% (SSD), and 50.0% (IBS-SSD) as was calculated through the scoring on the PHQ-15. When considering mean PHQ-15 scores, mean scores were highest amongst the SSD 17.8 +/- 5.27 (5.0–26.0), yet mean scores in the IBS sample 14.76 +/- 5.07 (4.0–24.0) and the IBS-SSD 14.92 +/- 6.27 (7.0–26.0) sample were also close to 15-an indicator of trend towards severity of somatization symptoms in participants. As expected, the sample of patients with the SSD diagnosis in our study had the highest percentage of participants with severe symptoms of somatization as compared to patients with IBS diagnosis and patients with a diagnosis of both IBS and SSD. However, IBS patients and patients with IBS and SSD still had high percentages of those with severe symptoms of somatization (>50%). This clearly reiterates the importance of identifying if IBS patients have a comorbidity of somatoform disorders. When considering overall SSRS scores means in each sample, means in the IBS sample 75.20 +/-19.59 (46–136), SSD sample 85.67+/-23.26 (44–127), and IBS-SSD sample 75.17+/-20.55 (52–131) were above 64 i.e., the average value for the stress-related somatization symptoms. Thus, considering if increasing somatization scores also increases stress-related somatization scores was evaluated through the linear bivariate regression. A positive relationship was found between the somatization symptomology and stress-related somatization symptomology. This means that increasing with increased somatization symptomology, stress-related somatization symptomology also increases. In our study, we believe that the sample that reported diagnosed comorbidity of SSD with IBS may be much higher in the overall population of IBS patients. This is because the reported somatization symptomology is not only in a large percentage of patients but also severe in nature. This estimation is synonymous with the fact that about half of patients with IBS report additional somatic and mental symptoms once they are interviewed [18, 24]. Similarly, with the increase in the severity of IBS, risk of comorbidities also increases [25]. In our study also, there were also more participants who either refused to complete the questionnaire or were not interested at all. Additionally, there was more male representation that may be explained by the fact that some of the females visiting the gastroenterology clinic for IBS diagnosis wanted to participate and share their story. However, their family members accompanying them to the clinic did not want them to participate in the study. This factor may have contributed to some extent in the female representation in the study sample.

Stress holds a significant value in affecting clinical outcomes in IBS patients [2]. PHQ scores indicate somatization symptom burden and SSRS scores indicate somatization symptom burden due to stress. Therefore, it was necessary to assess if the somatization symptom burden increases with stress burden in the subset of patients with IBS. Tools to assess stress-related somatic symptoms have rarely been developed. The Stress Response Inventory (SRI), which includes emotional, somatic, cognitive and behavioral stress responses, includes only a limited

number of items on the somatic symptoms [26]. We decided to use the SSRS instrument for our study. SSRS is highly reliable and valid, and that it can be effectively utilized as a measure for research of the somatic symptoms related to stress [23]. In the samples of patients with IBS, SSRS was correlated with PHQ in a linear fashion. The relationship of SSRS with PHQ in IBS patients has not been documented before.

Age seems to be an important factor in understanding the relationship of stress related somatic complaints and somatic burden in IBS patients. PHQ scores and SSRS were both highest for age group 60–69 years amongst IBS patients. It has been reported that the burden which elderly people with a somatoform condition experience is underlined by the findings that reported functional impairment is higher for those suffering from abridged somatization or pain compared to other respondents. Significantly increased rates of somatic symptom severity, as well as decreased rates of quality-of-life point towards marked impairments of somatoform conditions in the elderly [27]. To our knowledge, this phenomenon of high PHQ and SSRS scores in elderly patients with IBS has not been reported before. This also underscores the importance of incorporating psycho-social interventions for elderly patients with IBS, especially provided the increased acceptance of the potential role of "Acceptance and Commitment Therapy" (ACT) in the self-management of chronic illnesses [28]. ACT is based on the Commonsense Model of Self-Regulation (CSM) that explains the role of illness perceptions and representations in adhering to the self-management of chronic illnesses [29, 30]. Intuitively though, a clinician's inclination is to investigate the elderly with GI complaints mainly from an organic standpoint; it is equally important to seek or advise psychological help for this age group with gastrointestinal complaints, once a thorough work up has been undertaken.

Our study was more about the needs assessment of our priority population to plan and implement targeted interventions in future. However, this was not an interventional study. In future, interventional studies may be planned and implemented to evaluate the effectiveness of those interventions. Furthermore, what triggers the somatization symptoms in IBS patients is not clearly understood. Therefore, a study may be conducted to assess the psycho-social determinants of somatization in future. In our study, only education appeared to be an important variable responsible for differences in PHQ scores for the IBS sample. Additionally, longitudinal studies may be needed to assess cause-effect relationships for variables of interest relevant to somatization symptomology and stress-related somatization symptomology. We expected a much larger participation from the SSD sample. However, more than 30% of the SSD population refused to participate in the study. Therefore, in future additional steps may be taken to ensure larger participation.

## Conclusions

Stress related somatic symptoms are positively correlated with somatization complaints in IBS patients. Increased prevalence of somatization symptoms is most noticeable in the elderly. Somatization symptoms also differ by education levels. Targeted psycho-social interventions like ACT may help as a complementary intervention by considering the foundational differences by education to mitigate the negative effects of somatization in IBS patients.

## Supporting information

**S1 Dataset. Dataset used in this study for inferential statistics.** This is the data file for the data on Somatization symptomology and its association with stress in patients with irritable bowel syndrome (IBS).
(SAV)

**S2 Dataset. Dataset used in this study for inferential statistics.** This is the data file for the data on Somatization symptomology and its association with stress in patients with somatic symptom disorder (SSD).
(SAV)

**S3 Dataset. Dataset used in this study for inferential statistics.** This is the data file for the data on Somatization symptomology and its association with stress in patients with irritable bowel syndrome who have a comorbidity of somatic symptom disorder (IBS-SSD).
(SAV)

**S1 Table. Stress related somatization burden by age in IBS patients.** This is the table that represents the different PHQ scores in each age range for the IBS sample.
(DOCX)

## Author Contributions

**Conceptualization:** Saira Akhlaq, Nosheen Kazmi, Muslim Atiq.

**Data curation:** Saira Akhlaq, Sajawal Hussain, Ahmed Bajwa, Abdul Ahad, Muhammad Yaqoob Akhtar, Mohammad Rizwan.

**Formal analysis:** Saira Akhlaq, Sajawal Hussain.

**Investigation:** Saira Akhlaq, Nosheen Kazmi, Syed Murtaza Hassan Kazmi.

**Methodology:** Saira Akhlaq, Nosheen Kazmi.

**Project administration:** Saira Akhlaq, Muslim Atiq, Sajawal Hussain, Ahmed Bajwa, Abdul Ahad, Muhammad Yaqoob Akhtar.

**Resources:** Saira Akhlaq, Syed Murtaza Hassan Kazmi.

**Software:** Saira Akhlaq.

**Supervision:** Saira Akhlaq.

**Validation:** Saira Akhlaq, Kalsoom Akhlaq.

**Visualization:** Saira Akhlaq, Syed Murtaza Hassan Kazmi.

**Writing – original draft:** Saira Akhlaq.

**Writing – review & editing:** Saira Akhlaq, Muslim Atiq.

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
