## [Decision Letter · Decision Letter 0]

7 Jul 2024

PONE-D-24-25522Somatization Symptomology and Its Association with Stress in Patients with Irritable Bowel SyndromePLOS ONE

Dear Dr. Akhlaq,

Thank you for submitting your manuscript to PLOS ONE. After careful consideration, we feel that it has merit but does not fully meet PLOS ONE’s publication criteria as it currently stands. Therefore, we invite you to submit a revised version of the manuscript that addresses the points raised during the review process.

We look forward to receiving your revised manuscript.

Kind regards,

Muddsar Hameed

Academic Editor

PLOS ONE

Journal Requirements:

"Acknowledgments: Study is grant funded by the STMU Internal Grants with the Grant code 025-2023."

"Only S.A, N. K & M. A received funding for the administration of the project. Less than 50% of the APC may be covered by the funding agency. Funding agency is Shifa Tameer-e-Millat University Internal Reseach Grants Committee. Grant code has been provided under acknowledgment statement on the manuscript. No, the funders did not play any role in the study design, data collection and analysis, decision to publish, or preparation of the manuscript."

5. We note you have included a table to which you do not refer in the text of your manuscript. Please ensure that you refer to Table 5 in your text; if accepted, production will need this reference to link the reader to the Table.

Reviewers' comments:

Reviewer's Responses to Questions

**Comments to the Author**

1. Is the manuscript technically sound, and do the data support the conclusions?

Reviewer #1: Yes

Reviewer #2: Yes

2. Has the statistical analysis been performed appropriately and rigorously? 

Reviewer #1: Yes

Reviewer #2: Yes

3. Have the authors made all data underlying the findings in their manuscript fully available?

Reviewer #1: Yes

Reviewer #2: Yes

4. Is the manuscript presented in an intelligible fashion and written in standard English?

Reviewer #1: Yes

Reviewer #2: Yes

5. Review Comments to the Author

Reviewer #1: The article has several comments, including insufficient details in the abstract, an inadequate review of relevant literature in the introduction, lack of clarity in the methods section, incomplete presentation of results, and a conclusion that does not address limitations or future research directions. These issues need to be addressed to make the article more comprehensive, transparent, and informative.

Reviewer #2: The study "Somatization Symptomology and Its Association with Stress in Patients with Irritable Bowel Syndrome" addresses a relevant topic by exploring the overlap of somatization symptoms and the role of stress in IBS and SSD patients. However, the use of purposeful sampling introduces potential bias, and the small, heterogeneous sample size limits generalizability. The lack of demographic details, validation of measurement tools, and comprehensive statistical analysis weakens the findings. The study's conclusions are speculative without causal evidence from longitudinal or experimental designs. To enhance validity, future research should employ random sampling, larger sample sizes, validated tools, detailed statistical methods, and a thorough literature review, while ensuring ethical compliance.

6. PLOS authors have the option to publish the peer review history of their article (what does this mean?). If published, this will include your full peer review and any attached files.

Reviewer #1: No

Reviewer #2: No

---

## [Author Response · Author response to Decision Letter 0]

23 Aug 2024

Reviewer 1

Response: Abstract has been updated. We have included frequencies and percentages of participants now as initially how the data was entered in the SPSS spreadsheet; it was coded rather than entering original ages of each participant. Participants in the age range 18-29 years is coded as 1, 30-39 years as category 2, 40-49 years as category 3, 50-59 as category 4, 60-69 as category 5. Similarly, gender distribution has been included now. There were more males than females in all three samples. 

Response: Linear regression used was a simple linear regression. In other words, it was linear bivariate regression. It has been included in the manuscript now especially under sample size calculation. 

Response: p-values were reported previously. However, confidence intervals have also been included now especially for the reporting of linear regression results in the main body of manuscript as well as the abstract.

Introduction

Response: A full new introductory paragraph has now been added and addition of few sentences in the previous content to elaborate and logistically explain the relevance of current study by building on previous research.

Response- Physiological and psychological mechanisms that link stress and somatization in IBS patients has now been explained in the first paragraph of the manuscript especially by discussing the hypothalamus-pituitary-adrenal axis and the related concepts. Furthermore, this manuscript is one of the three manuscripts that are likely to come out from the entire grant-funded research. The entire grant funded project is based on the Common-Sense Model of Self-Regulation and future aim is to evaluate the effectiveness of ACT based on the CSM in the self-management of somatization in IBS, SSD, and IBS-SSD. However, that discussion will be done one we publish our findings on the third variable that is coping in association the pathophysiology in these samples. That manuscript will be published in another journal. 

Response- No previous study has previously discussed stress, somatization and IBS together. However, some studies discussed these concepts separately. Those studies have now been cited in the manuscript. 

Methods

Response- This was previously also explained. However, now we have explained why we chose purposeful. Patients already had the diagnosis from trained physicians either from gastroenterology clinic or psychiatry clinic. Rome IV criteria was used for IBS diagnosis and DSM V criteria was used for SSD diagnosis. This information was also part of the original manuscript under materials and methods. 

Response- Power analysis was also done before and reported in the original manuscript. However, the way findings were reported created a confusion. Therefore, information has been re-worded. IBS patients, SSD patients, and IBS-SSD patients have been considered separately for analysis. Therefore, G-Power was used to estimate sample sizes needed for generalizability based on statistical tests used. IBS sample in our study was the largest its findings are generalizable to larger populations. It included 67 participants and maximum 55 participants were needed for running ANOVA and 21 participants were only needed for linear regressions. Our sample is larger than both the numbers needed for generalizability. 

Response- I have calculated the Cronbach alpha for both the tools in the IBS sample now and the reliability is greater than 0.9 for both PHQ-15 and SSRS. I have also updated the manuscript. For the validity, considering the construct validity, both the tools are appropriate as they exactly measure the constructs that were needed in the study. This is one of the reasons that we did not use other tools available and accessible. A small part on this content has been included in the discussion section. 

Response- Yes, the tools were validated in previous research. This is the content in the original manuscript that explains the validity of SSRS (reference number 23) in the patients with somatization symptoms and the validity of PHQ-15 has been explained by citing reference number 22.

“Convergent validity of the SSRS was calculated by correlating the SSRC scale scores with the somatization sub-scale scores and other sub-scale scores in the Korean version of the SCL-90. and Discriminant validity of the SSRS was calculated by comparing the sub-scales of the healthy study participants with the patient study participants.” PHQ-15 is a shortened version of PHQ that is excessively being used in clinical settings. 

• Statistical Analysis:

Response-Descriptive statistics includes frequencies, and percentages of demographic variables, and means of SSRS scores and PHQ-15 scores. Omnibus, one-way ANOVA was conducted to assess the differences in PHQ scores by education and linear bivariate regression was conducted to assess changes in SSRS scores with the change in PHQ-15 scores. 

There was no missing data, and assumptions like test of normality, Durbin-Watson statistics, and Collinearity diagnostics were assessed. 

Results

• Demographic Information:

Response- This section has now been re-written. 

o Recommendations:

Response- Done.

• Presentation of Results:

Response- Done

Include regression coefficients, confidence intervals, and p-values for the linear regression analysis. This will provide a clearer understanding of the relationship between stress and somatization symptoms.

Response- Done

• Interpretation of Findings:

o Recommendations:

• Response- This was included in the original manuscript also. It is in the discussion section. It is the second-last paragraph under discussion.

Response- This point has now been discussed in the first paragraph by citing a reference to a study by Benette. However, a complete discussion on this point can be done after this study as the original plan was to first do a survey on the needs assessment of our priority population. Then after the completion of this study, assess if stress can be used as a moderator by assessing through the Path analysis model. It would be a good idea to consider the severity of IBS symptoms as a moderator variable instead. However, this is beyond the scope of the current study. Irrespectively, severity of ibs symptoms is important and has been discussed in relation to stress and somatization in IBS sample. 

Conclusion

Response- This section has now been included.

• Recommendations:

Response- Done

o Suggest directions for future research. 

Response- Done.

Reviewer 2

• Stratify Participant Groups:

Response- Acknowledged. Sorry about the confusion created through the original manuscript. Analysis have been done separately. These were three different samples from three different populations. Only IBS sample was large enough for generalizability. However, as the analysis were done on all the samples and were significant, it was appropriate to report the findings of all the samples. The wording “sub-groups” was inaccurate and now has been replaced with “sample” or samples as needed.

• Detail Statistical Methods:

Response- Done in the manuscript now.

• Enhance Results Presentation:

Response- Done now. 

Contextualize Findings with Literature:

Response- Additional information has now been included.

---

## [Editor Report · Decision Letter 1]

17 Sep 2024

Somatization Symptomology and Its Association with Stress in Patients with Irritable Bowel Syndrome

PONE-D-24-25522R1

Dear Saira AKlaq,

We’re pleased to inform you that your manuscript has been judged scientifically suitable for publication and will be formally accepted for publication once it meets all outstanding technical requirements.

Kind regards,

Muddsar Hameed

Academic Editor

PLOS ONE
---

## [Editor Report · Acceptance letter]

17 Oct 2024

PONE-D-24-25522R1 

PLOS ONE

Dear Dr. Akhlaq, 

I'm pleased to inform you that your manuscript has been deemed suitable for publication in PLOS ONE. Congratulations! Your manuscript is now being handed over to our production team.

Kind regards, 

on behalf of

Dr. Muddsar Hameed 

Academic Editor

PLOS ONE